# The Dance of Pauses in Poetry Declamation

Plinio A. Barbosa 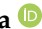

Department of Linguistics, University of Campinas, Campinas 13083-859, Brazil; pbarbosa@unicamp.br

**Abstract:** In poetry declamation, the appropriate use of prosody to cause pleasure is essential. Among the prosodic parameters, pause is one of the most effective to engage the listeners and provide them with a pleasant experience. The declamation of three poems in two varieties of Portuguese by ten Brazilian Portuguese (BP) speakers and ten European Portuguese (EP) speakers, balanced for gender, was used as a corpus for evaluating the degree of pleasantness by listeners from the same language variety. The distributions of pause duration and inter-pause interval (IPI) both varied greatly across the subjects, being the main source of variability and strongly right-tailed. The evaluation of the degree of pleasantness revealed that pause duration predicts degree of pleasantness in EP, whereas IPI predicts degree of pleasantness in BP. Reciters perform a kind of complex "dance", where sonority between pauses is favored in BP and pause duration in EP.

**Keywords:** poem declamation; pause; pleasantness; acoustic phonetics; Portuguese

## 1. Introduction

A certain amount of work on the roles and acoustic correlates of pauses in speech can be found in the literature (see Duez 1993; Zellner 1994; Swerts 1997; Fletcher 2010; Tyler 2013; Mittmann and Barbosa 2016). According to these studies and surveys, it is undeniable that pausing contributes to organizing the speech chain into smaller chunks for the sake of both the speaker and the listener. This cognitive function of segmentation has been investigated by many authors since the 1960s (Goldman-Eisler 1961a, 1968, 1972; Ferreira 1993) and is connected to methods for inferring the structured hierarchy of prosodic grouping from patterns of silent pauses, such as the method proposed by Grosjean and colleagues (Grosjean et al. 1979; Gee and Grosjean 1983; Monnin and Grosjean 1993).

Appropriate pause position and duration are important parameters for comprehension, inspiring speech synthesis prosodic models that integrate pause generation (see Fujio et al. 1997; Barbosa and Bailly 1997). Specifically, pause duration can vary considerably across speaking styles and languages: longer pauses are found in political interviews vs. news announcements (Strangert 2005) in Swedish; longer pauses in narratives vs. conversations (Gyarmathy and Horváth 2019) in Hungarian; longer pauses in spontaneous vs. read speech in six languages, with highlighted aspects of cross-linguistic variation (Campione and Véronis 2002); longer post-message pauses in Barack Obama's climate change address to the United Nations vs. mid-message pauses (Niebuhr and Banzina 2022); longer pauses and higher rates of pause in religious and political discourses vs. broadcast news and interviews in Brazilian Portuguese (Castro 2008); a higher variability in pause production is found across speakers and for different speaking rates (Werner et al. 2022) in German. Some studies recognize that part of this variability is due to different causes and changes in behavior related to distinctive cognitive loads (see Goldman-Eisler 1961b; Kirsner et al. 2002; Merlo and Barbosa 2010), which also explain differences across speaking styles.

As for the esthetical use of pauses in theater and poetry declamation, their role in provoking expectation by creating dramatic effect is well known by actors, reciters and narrators (see Buraa 2019 for the effects of pause in the Theater of the Absurd). Regarding poetry declamation, Postarnak et al. (2020) showed that the realization of pauses in

places not predictable from syntax increased perceptions of pleasure, whereas Byers (1979) proposed that the temporal aspects of poetry declamation are characterized by slower speech rates, more frequent pauses, and short tone units.

By investigating the perceptual effects of the production choices of one male and one female professional speaker reciting the poem "Soneto da Fidelidade" ("Sonnet of Fidelity") by Vinícius de Morais in Brazilian Portuguese, Madureira (2008) showed that a whispery voice, F0 narrow range, and a great number of silent pauses were the prosodic parameters used by the female speaker to express negative affects, whereas the male speaker employed more varied intonation patterns to sound more expressive in order to provoke a sensation of liveliness.

The present work builds on our previous research on the acoustics and perception of poetry declamation (Barbosa 2022), which investigated the link between the appreciation of wellbeing and pleasantness, and several prosodic parameters including silent pauses. Because the adjective "agradável" (pleasant) has general use in Portuguese for appraisal of external events, no particular instructions were given to the participants on how to interpret the term. This experimental approach is often found in the literature that considers "pleasantness" in speech (see, for instance, Carlsen et al. 2018; Weiss and Burkhardt 2012). An inter-rater reliability test evaluated the coherence between the listeners' responses, and is described in the Results section.

We showed that the ways that reciters use pauses are important for causing pleasure: declamations with longer pause durations and lower pausing rates were considered more pleasant with effect sizes superior to the other prosodic parameters for explaining the evaluated degree of pleasantness. The participants were ten Brazilian lay reciters and ten Portuguese lay reciters. They were evaluated by ten listeners of their own language variety for the declamation of the poem "Quando vier a primavera" ("When Spring comes") from Alberto Caeiro, one of Fernando Pessoa's heteronyms. The fact that the poem addresses death through the passing of the seasons raised issues concerning the possible bias effect of its semantic content on the listeners' judgments.

Therefore, the work presented here investigates the production and perception aspects of the declamation of three other poems, where valence (positive and negative) and poet (Alberto Caeiro, a Portuguese poet, and Adélia Prado, a Brazilian poet) were contrasted. The present analyses focus on pause duration and rate, due to their significant relevance to pleasantness as reported in the previous investigation.

Based on the reviewed literature, our previous findings with single poem declamation, and the intuition that poetic declamation is a genre on its own that does not depend on the poem's content, the main hypotheses of the present study are:

**Hypothesis 1 (H1).** *The distributions of silent pause duration and rate are not dependent on the valence of the poem.*

**Hypothesis 2 (H2).** *The distributions of silent pause duration and rate are not dependent on the poet (covariate with the general theme and genre).*

**Hypothesis 3 (H3).** *There are no differences in the distribution of silent pause duration according to the gender of the reciter.*

**Hypothesis 4 (H4).** *There are no differences in the distribution of pause rate according to the gender of the reciter.*

**Hypothesis 5 (H5).** *The main source of variability in terms of pause-related parameters is inter-reciter variation.*

**Hypothesis 6 (H6).** *Declamations with longer pauses are considered more pleasant.*

**Hypothesis 7 (H7).** *Declamations with lower rates of pauses are considered more pleasant.*

## 2. Materials and Methods

The PROS-POIESIS corpus started with the declamation of the 155-word poem "Quando vier a primavera" ("When Spring comes") by Alberto Caeiro by ten Brazilian speakers and ten Portuguese speakers, of balanced genders, who recited it in their own varieties, i.e., Brazilian Portuguese (henceforth BP) or European Portuguese (henceforth EP). This corpus was supplemented with the declamation of three other poems in Portuguese by another set of ten Brazilian and ten Portuguese speakers: "O amor é uma companhia" ("Love is a Company", 111 words) by Alberto Caeiro, "Momento" ("Moment", 108 words), and "Tempo" ("Time", 71 words), both by Adélia Prado. They were chosen as shorter than the poem "Quando vier a primavera", for the sake of allowing the realization of the perception task described here in less than 30 minutes. These new declamations are referred here as the Current Set. When necessary, analyses include the declamations of the four poems. In these cases, the set of declamations is referred to as the Whole Set. All poems are found in Appendix A, for the reader's convenience.

Due to the reference to death in "Quando vier a primavera", its valence was labeled as negative, in contrast to the positive valence of the second poem by the same poet, "O amor é uma companhia", where the theme is the poet's joy in loving someone. Considering the same contrast of valence, the two poems by Adélia Prado contrast the positive valence of "Tempo" with the negative valence of "Momento".

Following the choices in the previous study, the new set of speakers included no-one with any professional voice training and all were between 25 and 50 years of age to avoid the effects of vocal aging (Stathopoulos et al. 2011) that usually affect melodic tone, voice quality, and temporal parameters such as rates of speech and pauses. The reason for not including the same speakers from the first study was simply the non-availability of the same individuals for the task of reciting the three new poems, either in Brazil or in Portugal.

Due to the COVID-19 pandemic, the participants themselves used the Easy Voice app on their own cell phones to make all the recordings. Because this app allows selection between different codifications, instructions were given to record all audio files in PCM format (WAV) at a sampling rate of 48 kHz. The author, who is a trained phonetician, further evaluated all audio files. The recordings were resampled at 16 kHz and standardized to the same maximum intensity of 65 dB.

Recruitment was based in the traditional friend-of-a-friend approach in sociolinguistics, started by the friends of the author in Brazil and the friends of a Portuguese colleague, Ana Rita Valente (University of Aveiro) in Portugal, in order to reach 20 participants and include both varieties, considering both speakers and listeners.

### 2.1. Acoustic Parameters

The Prosody Descriptor Extractor script for Praat (Boersma and Weenink 2021) implemented by the author (Barbosa 2020) was used to extract 22 prosodic–acoustic parameters from a verse-based segmentation of the poems (see Appendix A for the display of the verses in each poem, the same display sent to the participants). From these parameters, only the two related to silent pauses are considered in the present work: pause duration and rate. The declamation of each the three poems under investigation here lasted from 31 to 61 s. An example of the segmentation can be seen in Figure 1. Pauses were marked manually, delimited by the cessation of speech at the left boundary and resuming of speech to the right, both guided by broadband spectrograms. A particular pause may contain audible intakes, depending on the reciter.

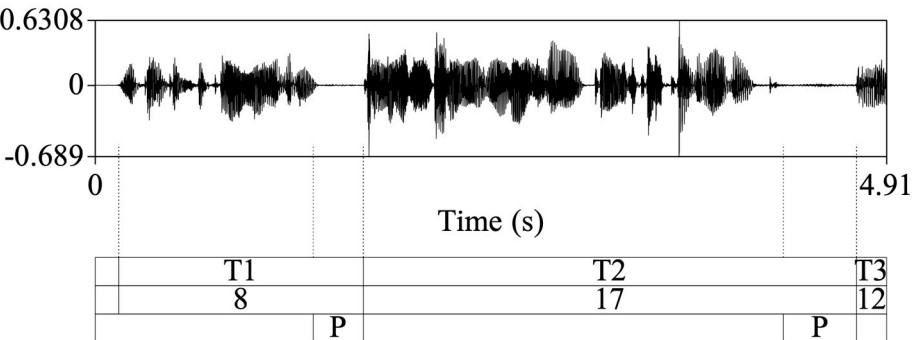

**Figure 1.** Example of the verse-wise segmentation (top tier, verses T1, T2, and start of T3), the number of phonological syllables in the corresponding verse (middle tier), and the silent pause intervals (bottom tier) marked with "P".

### 2.2. Perceptual Task

Two perception tests were carried out: one Likert-scale test for evaluating the degree of pleasantness, which was carried out by ten Brazilian listeners who rated the declamations of all Brazilian reciters, and one Likert-scale test for evaluating the degree of pleasantness, carried out by ten Portuguese listeners who rated the declamations of all Portuguese reciters. For that task, each participant listened to the entire declamation of all reciters of his or her variety, which totaled 30 declamations to be evaluated per variety (10 reciters × 3 poems). The order of the poems to be evaluated was randomized in each variety of Portuguese and maintained the same for all listeners of the respective variety. Due to the nature of the task, an esthetical evaluation of a relatively long stretch of sound, we do not think that the order of the recitations could strongly affect the results.

The scale for pleasantness evaluation varied in five degrees from "very unpleasant" ("muito desagradável" in Portuguese, degree 1) to "very pleasant" ("muito agradável" in Portuguese, degree 5) with the neutral response ("neutro" in Portuguese) at degree 3. Because these labels have general use in Portuguese for evaluating external events as pleasant ("agradável"), no particular instructions were given to the participants on how to interpret the terminology.

For the perceptual tests, the listeners were selected from the same age range as the reciters, that is, between 25 and 50 years of age, to avoid effects of differences in evaluation by younger or older age groups.

### 2.3. Statistical Analyses

To test our hypotheses, five kinds of statistical models were run in R (R Development Core Team 2013) software. Due to the non-normality of pause duration and rate distributions, a characteristic also found by other investigators (see Campione and Véronis 2002; Kirsner et al. 2002), a set of Scheirer–Ray–Hare (SRH) tests, the non-parametric equivalent of a Two-Way ANOVA test, was used for evaluating mean differences in terms of pause duration and inter-pause-intervals (IPI) according to the factors POET (Adélia Prado vs. Alberto Caeiro) and VALENCE (positive vs. negative). Working with IPI is equivalent to working with pause rate, because the inverse of mean IPI is the rate of pauses. For the first hypothesis, these models were split for gender and variety, accordingly.

Two mixed models per variety, to explain the variance of pause duration and IPI, were built to evaluate the strength of two fixed factors, POET and VALENCE, and a random factor, participant. These models revealed that the main source of variability was the participant.

A third set of statistical models included non-parametric Wilcox multicomparison variance tests followed by corresponding Duncan tests, to evaluate groupings of participants in terms of behavior for both pause duration and IPI.

The fourth statistical model was logistic regression. Two logistic regression models, one in each variety, corresponding respectively to the two Likert-scale tests presented in the

previous section, were retained predicting the degree of pleasantness from two predictor variables: pause duration and IPI. For doing so, the scores from 1 to 5 were respectively transformed to 0 to 100% in intervals of 25%, allowing the use of a logistic model, to test for predicting proportions. The quasi-binomial family in the R glm function was used for prediction when there were important differences between degrees of freedom and amount of deviance, according to statistical theory (Roback and Legler 2021, chp. 6). When this was not the case, the binomial family was used. Nagelkerke's pseudo-correlation measures were employed to evaluate the degree of explained variance of these models, which are measures of effect size.

The fifth and final kind of statistical model was an inter-rater reliability test computed by Krippendorff's alpha (Gittinger et al. 2022) to assess how close the evaluations of pleasantness for the 30 recitations were in each variety of Portuguese.

In all tests, a 0.05 level of significance was used for decisions regarding statistical differences.

## 3. Results

In order to evaluate the first two study hypotheses, which correspond to the role of valence and poet (consequently, specific content) in pause distributions, a general description of the participants' declamation is necessary. The first two sets of statistical testing used the Whole Set. The reason for not using the Whole Set for the third and fourth sets of statistical tests was the difference in terms of participants between that set and the Current Set. The statistical models that included the different participants of the Whole Set assumed that these groups behaved homogeneously in terms of the variables under study.

### 3.1. Pause Production

Considering the declamations of the Whole Set, for all participants, longer pauses appeared after the end of a verse. However, not all verse endings triggered a pause, due to the use of enjambement, that is, the continuing of a sentence from one verse into the start of the next verse. Some participants preferred to respect the arrangement of the verses and paused at the verse end, even if it disrupted the syntactic structure of the sentence, whereas others preferred to respect the syntax of the sentence and did not pause between the verses. For instance, Brazilian male participant M3 paused, indicated by #, between the verses, here separated by a slash, of the "Momento" poem in the sequence "constituindo o mundo pra mim, anteparo/# para o que foi um acometimento", whereas Brazilian participant M5 did not pause at that point, connecting the sentence and pausing in the place corresponding to the comma: "constituindo o mundo pra mim, # anteparo/para o que foi um acometimento". This kind of difference in pause realization produced distinct pause durations and IPI distributions across reciters.

Four SRH models per variety were applied to the Whole Set for each gender and each dependent variable, with POET and VALENCE as factors. These models revealed no significant results for either pause duration or IPI in BP, meaning that, for that variety, neither the valence nor the poet produced different behaviors for the two genders. For EP, IPI was significantly lower when the valence was negative, for male ($p = 0.001$) and female ($p = 0.002$) reciters. In that variety, only male reciters exhibited a significant difference in pause duration for a single poem ($p = 0.01$). These results confirm that hypotheses H1 to H4 can be accepted only for BP.

For EP, the situation is more complex because the reciters behaved differently regarding valence and poet. Portuguese reciters produced a higher pause rate when reading the negative poems from either poet. For males, this distinction also applies for pause duration, with a mean value of 641 ms when reciting Alberto Caeiro's positive poem, distinct from both the negative poem by the same poet, with a mean of 482 ms, and from the positive poem by Adélia Prado, with a mean of 470 ms.

As an illustration, it can be observed in Figure 2 that for the IPI of Portuguese females reciting the poems by Adélia Prado, lower median values of the variable were associated with negative valence for four out of five reciters (F2EP the only exception).

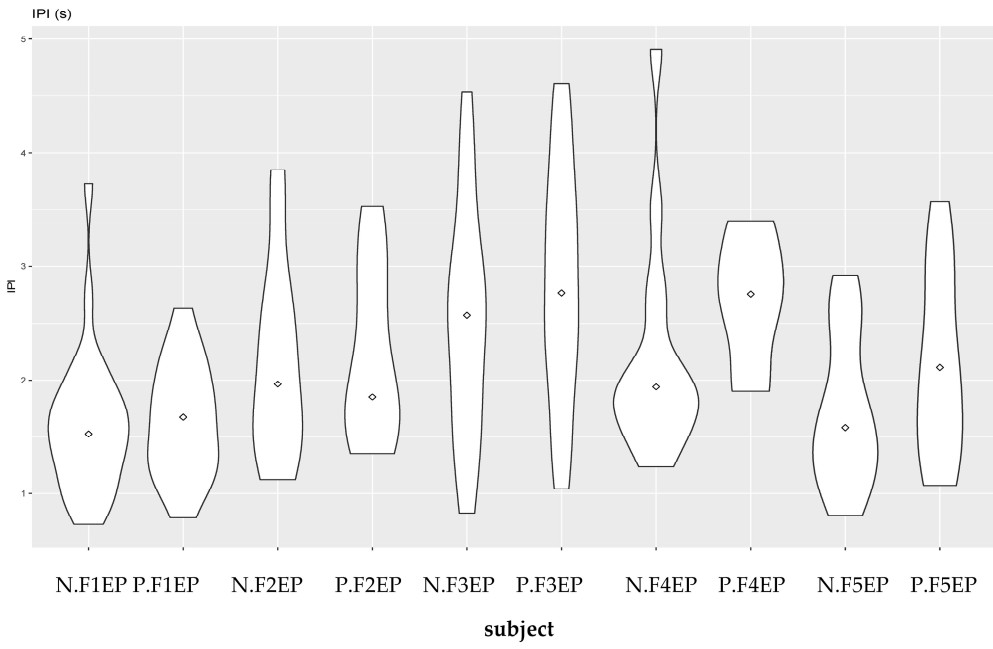

**Figure 2.** IPI (s) violin plots for the five Portuguese female participants of the Current Set. Valence is referred to as N(egative) or P(ositive) in the x-axis, with the speaker ID.

The strikingly similar gender-wise patterns of both pause duration and IPI in BP can respectively be seen in Figures 3 and 4, for the Brazilian participants of the Current Set. Note that these histograms reveal extended top tails, a characteristic of pause distributions previously discussed by Campione and Véronis (2002).

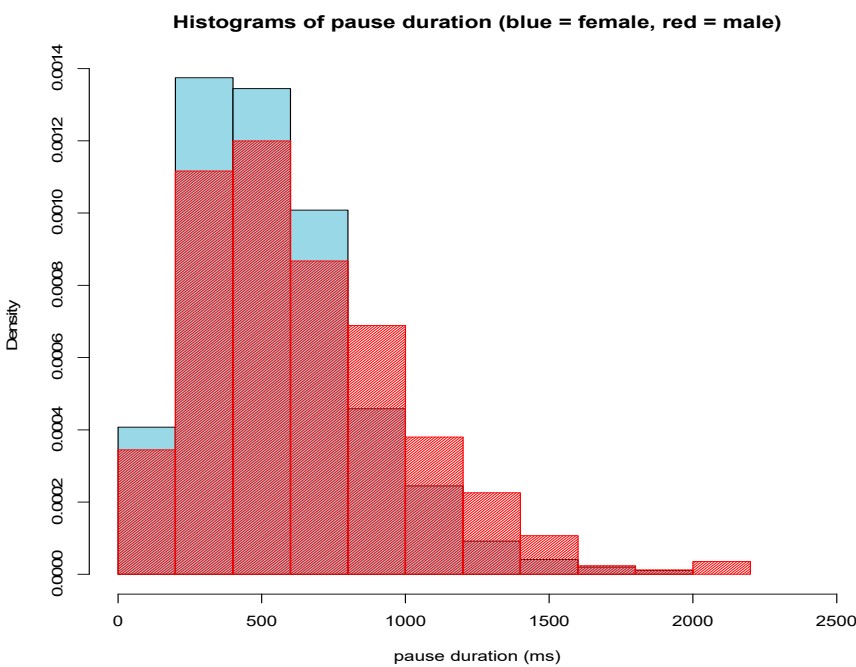

**Figure 3.** Superposed histograms for pause duration in milliseconds for females (blue) and males (red) for the Brazilian participants in the Current Set.

**Histograms of IPIs (blue = female, red = male)**

**Figure 4.** Superposed histograms for IPI (in seconds) for females (blue) and males (red), for the Brazilian participants in the Current Set.

Mixed models using the rank of each dependent variable (pause duration and IPI) as the predicted variables, POET and VALENCE as fixed-factors and PARTICIPANT as the random factor revealed that both fixed factors, when significant, explained only up to 4% of the variance of either pause duration or IPI, whereas PARTICIPANT explained between 5 and 12% of the total variance, considering the two varieties under study here. The latter values increased to an interval between 51 and 84%, when the mean values of IPI and pause duration replace the individual values for these two variables for each participant.

In that case, the participant is the main source of variability in relation to IPI or pause duration. Other parameters not limited to pauses, such as melodic and voice-quality parameters, are responsible for larger percentages of explained variance from the participants, but they are not under scrutiny here.

Having attested that the valence and poet have an impact on the Portuguese reciters, and answered the first four hypotheses, the following analyses considers the Current Set only, in order to highlight the differences across participants. When necessary, comparisons are drawn with the recitation of "Quando vier a primavera" in the previous study.

According to Wilcox and pairwise variance tests, all participants differed in pause duration and IPI mean and variance. Figures 5–8 provide violin plots that point to inter-reciter differences between the two varieties of Portuguese. In Figure 5, the strong top tails of reciters F2BP, F5BP, and M4BP can be easily seen. A Duncan test revealed three overlapping data sample sets of participants in increasing order of pause duration mean: set 1 formed by F5BP, set 2 formed by F2BP, F3BP, F4BP, M2BP, M3BP, and M5BP, and set 3 formed by F1BP, M1BP, and M4BP.

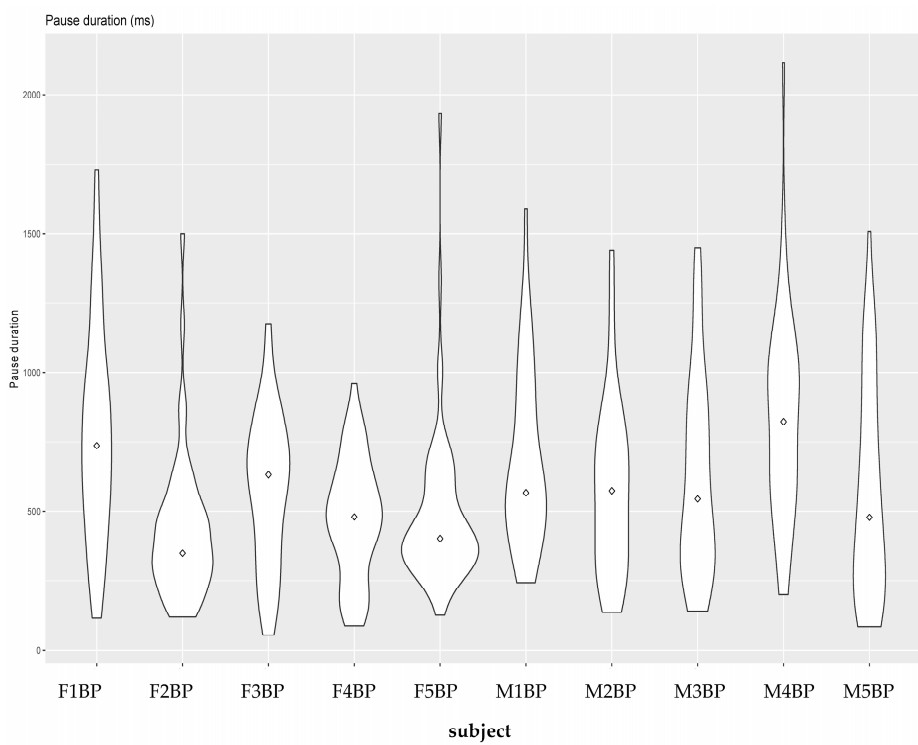

**Figure 5.** Pause duration (ms) violin plots for the ten Brazilian subjects of the Current Set. The five left-hand plots are from female reciters and the five right-hand plots from male reciters.

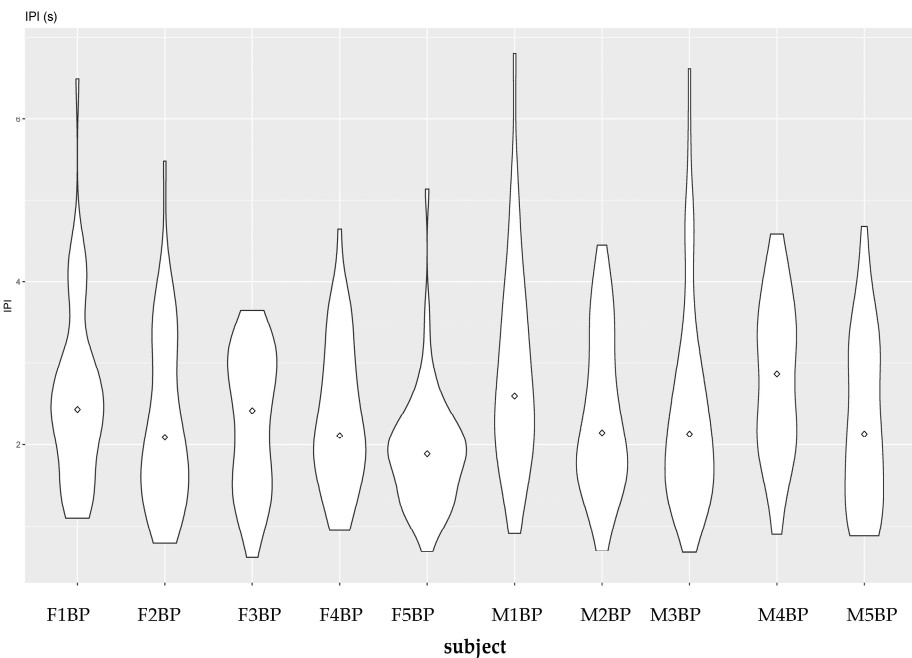

**Figure 6.** IPI (s) violin plots for the ten Brazilian subjects of the Current Set. The five left-hand plots are from female reciters and the five right-hand plots from male reciters.

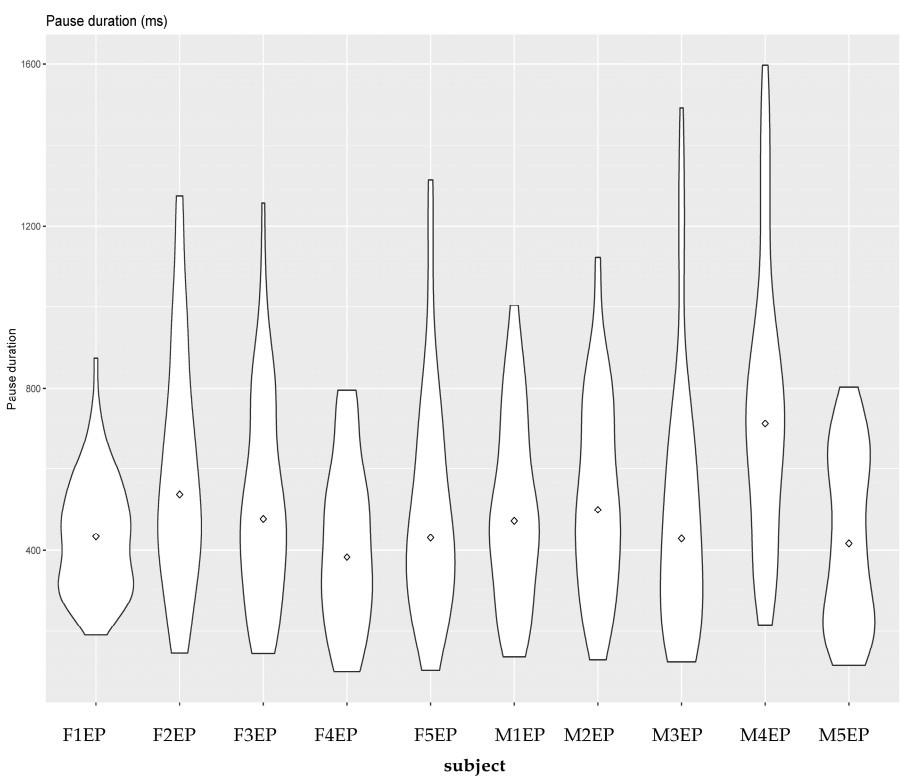

**Figure 7.** Pause duration (ms) violin plots for the ten Portuguese subjects of the Current Set. The five left-hand plots are from female reciters and the five right-hand plots from male reciters.

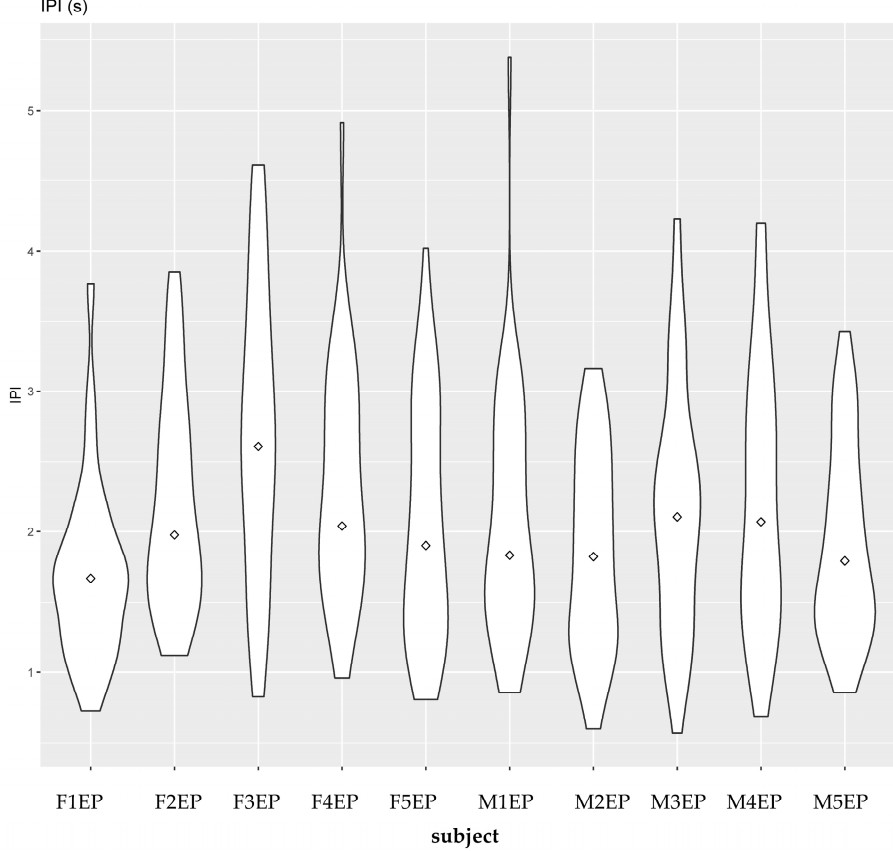

**Figure 8.** IPI (s) violin plots for the ten Portuguese subjects of the Current Set. The five left-hand plots are from female reciters and the five right-hand plots from male reciters.

Observe, in Figure 6, for IPI in BP, the strong top tails of reciters F1BP, F2BP, F5BP, M1BP, and M3BP. A Duncan test revealed the three overlapping data sample sets in increasing order of mean IPI: set 1 formed by F2BP, set 2 formed by F3BP, F4BP, F5BP, M2BP, and M5BP, and set 3 formed by F1BP, M1BP, and M3BP, and M4BP.

Observe, in Figure 7, for pause duration in EP, the long top tails of reciters F1EP, F2EP, F3EP, F5EP, M2EP, M3EP, and M4EP. A Duncan test revealed four overlapping data sample sets in decreasing order of mean pause duration: set 1 formed by M4EP, set 2 formed by F2EP, set 3 formed by M1EP, M2EP, M3EP, F3EP, and F5EP, and set 4 formed by F1EP, F4EP, and M5EP.

Finally, observe, in Figure 8, for IPI in EP, the strong top tails of reciters F1EP, F4EP, M1EP, M3EP, and M4EP. A Duncan test revealed four overlapping data sample sets in decreasing order of IPI mean: set 1 formed by F3EP, set 2 formed by F4EP, set 3 formed by M2EP, and M5EP, and set 4 formed by F1EP.

Table 1 shows the values for median pause duration, standard deviation, and Pearson skewness for the Brazilian reciters of the Current Set. With two exceptions (participants F3BP and F4BP), the values for skewness are right-tailed, pointing to the use of extra-long pauses, which in all cases were found at the ends of verses. Median values ranged from 350 to 823 ms, whereas standard deviations ranged from 213 to 382 ms.

**Table 1.** Median (and standard deviation) in ms, and Pearson skewness of pause duration for the Brazilian participants of the Current-Set.

| Participant | Median (Standard Deviation) | Skewness |
|---|---|---|
| F1BP | 736 (382) | 0.48 |
| F2BP | 350 (264) | 2.00 |
| F3BP | 630 (270) | −0.04 |
| F4BP | 479 (213) | 0.03 |
| F5BP | 403 (283) | 2.91 |
| M1BP | 567 (319) | 0.90 |
| M2BP | 574 (310) | 0.78 |
| M3BP | 545 (352) | 0.68 |
| M4BP | 823 (371) | 0.81 |
| M5BP | 478 (381) | 0.65 |

Table 2 shows the values for IPI median, standard deviation and Pearson skewness for the Brazilian reciters of the Current Set. With two exceptions (participants F3BP and M4BP), the values for skewness are right-tailed, pointing to lower rates of pausing, which depend on the extension of the verses, which vary for the three poems in the Current Set. Median values ranged from circa 2 to 3 s, corresponding to pause rates of 30 or 20 pauses per minute, respectively. Standard deviations ranged from 0.7 to 1.3 s.

**Table 2.** Median (and standard deviation) in s, and Pearson skewness of IPI for the Brazilian participants of the Current Set.

| Participant | Median (Standard Deviation) | Skewness |
|---|---|---|
| F1BP | 2.4 (1.1) | 1.13 |
| F2BP | 2.1 (1.1) | 0.84 |
| F3BP | 2.4 (0.9) | −0.21 |
| F4BP | 2.1 (0.9) | 0.49 |
| F5BP | 1.9 (0.7) | 1.70 |
| M1BP | 2.6 (1.3) | 1.03 |
| M2BP | 2.1 (1.0) | 0.45 |
| M3BP | 2.1 (1.3) | 1.39 |
| M4BP | 2.9 (1.0) | −0.06 |
| M5BP | 2.1 (1.0) | 0.36 |

Table 3 shows the values for pause duration median, standard deviation and Pearson skewness for the Portuguese reciters of the Current Set. The values for skewness are right-tailed, pointing to the use of long pauses, also found at the end of verses. Median values ranged from 383 to 715 ms, whereas standard deviations ranged from 142 to 343 ms.

**Table 3.** Median (and standard deviation) in ms, and Pearson skewness of pause duration for the Portuguese participants of the Current Set.

| Participant | Median (Standard Deviation) | Skewness |
|---|---|---|
| F1EP | 433 (142) | 0.59 |
| F2EP | 536 (289) | 0.60 |
| F3EP | 476 (257) | 0.66 |
| F4EP | 383 (202) | 0.29 |
| F5EP | 430 (277) | 1.04 |
| M1EP | 472 (227) | 0.36 |
| M2EP | 499 (240) | 0.40 |
| M3EP | 428 (324) | 1.21 |
| M4EP | 715 (343) | 0.65 |
| M5EP | 416 (211) | 0.15 |

Table 4 shows the values for median IPI, standard deviation, and Pearson skewness for the Portuguese reciters of the Current Set. The values for skewness are right-tailed, pointing to the non-frequent use of long pauses, coinciding with the end of verses. Median values ranged from 1.7 to 2.6 s, corresponding to a pause rate of 35 and 23 pauses per minute, respectively. Standard deviations ranged from 0.6 to 1 s. The pause rate w therefore higher than for BP.

**Table 4.** Median (and standard deviation) in s, and Pearson skewness of IPI for the Portuguese participants of the Current Set.

| Participant | Median (Standard Deviation) | Skewness |
|---|---|---|
| F1EP | 1.7 (0.6) | 1.24 |
| F2EP | 2.0 (0.8) | 0.63 |
| F3EP | 2.6 (1.0) | 0.12 |
| F4EP | 2.0 (0.8) | 0.84 |
| F5EP | 1.9 (0.8) | 0.41 |
| M1EP | 1.8 (0.9) | 1.34 |
| M2EP | 1.8 (0.7) | 0.17 |
| M3EP | 2.1 (0.9) | 0.44 |
| M4EP | 2.1 (0.9) | 0.45 |
| M5EP | 1.8 (0.7) | 0.57 |

*3.2. Prediction of Pleasantness from Pause*

Among the models to predict pleasantness from pause duration and IPI, the final logistic regression model for BP retained only IPI as a significant predictor (intercept a = 0.4, with t = 5.676, $p = 2.4 \times 10^{-8}$; IPI slope = 0.08, with t = 2.457, $p = 0.01$), explaining 50% of the variance (Negelkerke pseudo-correlation). The higher the IPI, the higher was the reported degree of pleasantness. This increase of pleasantness with IPI suggests that Brazilian listeners considered more pleasant the recitations with longer stretches of sound. A new logistic regression model with the duration of the sound between pauses in the recitation confirmed this assumption: intercept a = 0.48, with t = 6.81, $p = 3.0 \times 10^{-11}$; durSound slope = 0.09, with t = 2.606, $p = 0.009$. The 95% confidence interval of the duration stretches of sound between pauses is between 0.54 and 3.60 s.

The Krippendorff's alpha for the responses of BP listeners was 0.28, and that for EP listeners was 0.58, both values being significant. In this kind of task, the reliability is usually below 0.50 (see work on judging emotions in speech, for instance, Alm and Sproat 2005).

The most important thing to report is that EP listeners were more coherent in their responses than BP listeners, which is a matter for further investigation.

Figure 9 shows the distributions of attribution of degree of pleasantness by Brazilian listeners for the Brazilian reciters and poems (the number after the poet's abbreviation indicates valence: 2 is negative and 1 is positive. AP = Adélia Prado, and CA = Alberto Caeiro).

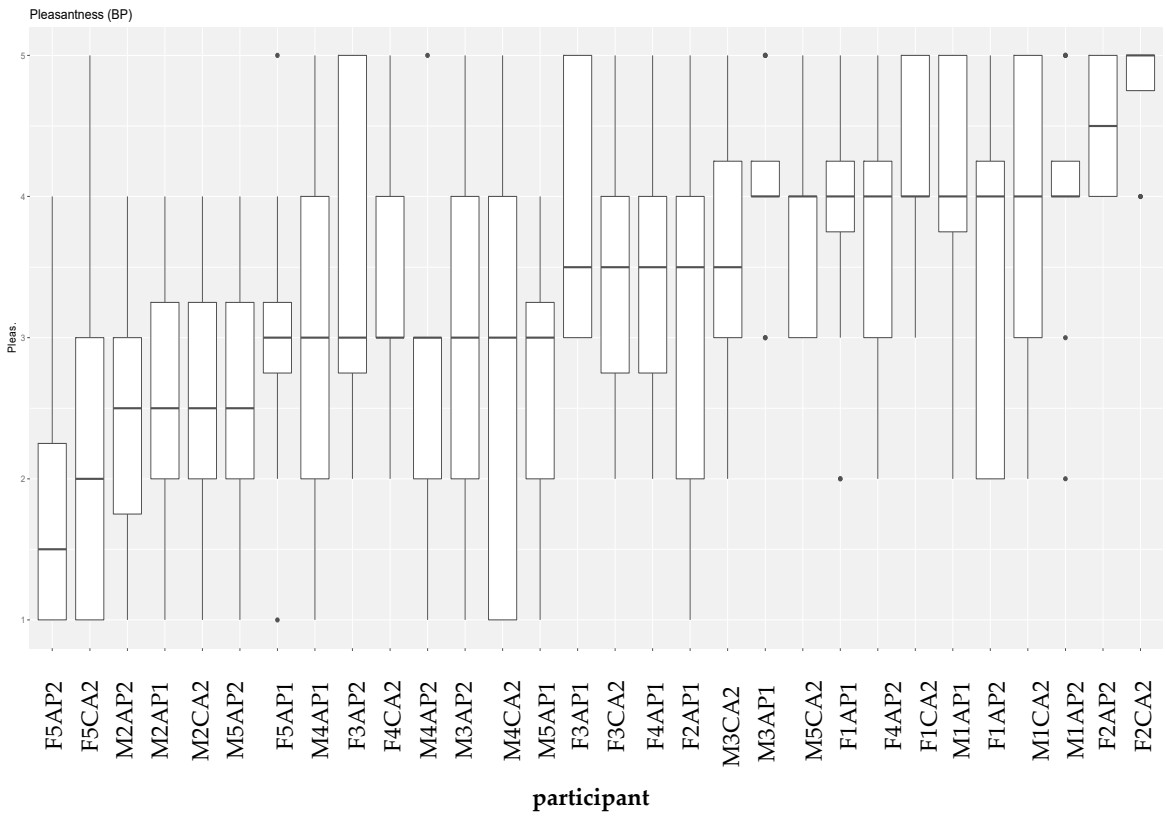

**Figure 9.** Pleasantness degree boxplots for BP declamations for each reciter and poem.

Figure 10 shows how pleasantness increases with the increasing of IPI (see median position). The median degree of 4.5 is an exception, which refers to the evaluation of the pleasantness of the declamation of the negative-valence Adélia Prado poem by BP female reciter F2. We invite the reader to listen to this declamation in the repository: a high melodic variation certainly explains the pleasantness attributed to the speaker's declamation, despite the unexpected lower value of IPI, meaning a higher rate of pauses per minute.

Regarding pleasantness prediction in EP, the final logistic regression model retained only pause duration for male reciters as a significant predictor (intercept a = −0.4, with t = −2.77, *p* = 0.006; pause duration slope = 0.001, with t = 4.81, *p* = $2.6 \times 10^{-6}$), explaining 72% of the variance (Negelkerke pseudo-correlation). This finding confirms what was found in the previous study for a single poem, "Quando vier a primavera", where EP differed from BP precisely because IPI was not a predictor for the former and was a predictor for the latter.

Figure 11 represents the increase in pleasantness degree attributed by the Portuguese listeners with increased pause duration by the male EP reciters. This increase is clear from degree 2.0 onwards.

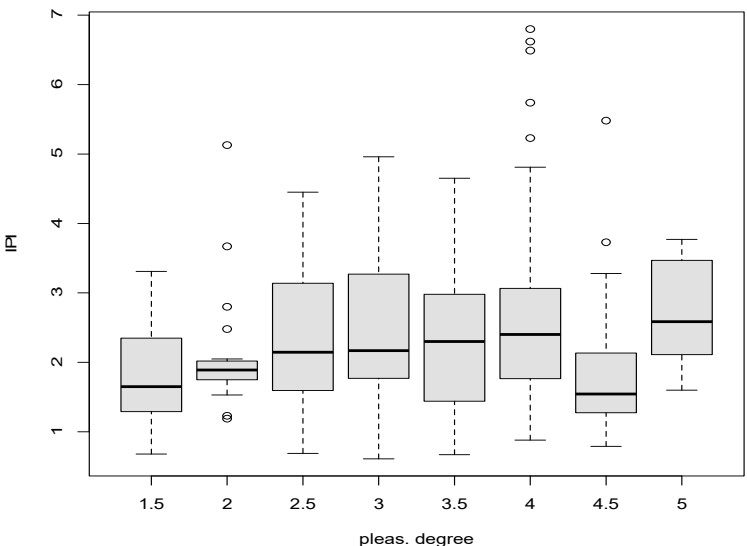

**Figure 10.** Pleasantness degree attributed by Brazilian listeners to the BP declamations, according to IPI (s). The circles outside the boxes are outliers.

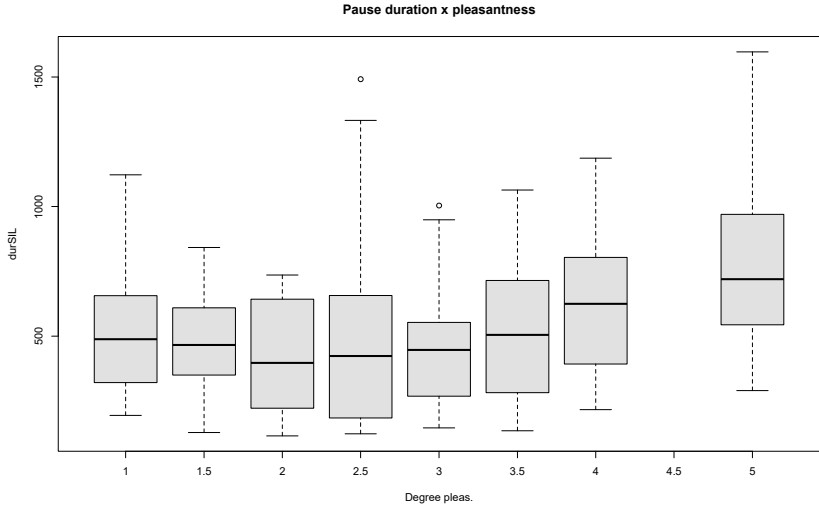

**Figure 11.** Pleasantness degree attributed by Portuguese listeners to the EP declamations, according to pause duration (ms). The circles outside the boxes are outliers.

## 4. Discussion

The first four hypotheses raised in the Introduction concern the dependence of pause duration and rate on poet, valence, and gender. The results presented in the Results section confirm that these hypotheses are true for BP, but not for EP. Within that variety, differences were found for the three factors: male reciters produced longer pauses when reciting the positive poem by Alberto Caeiro; both genders had higher pause rates when reciting the negative poems, which were also the longer ones (see Appendix A).

The fact that participants with high positive values for IPI skewness, such as F1BP, F2BP, and M1BP (see Figures 6 and 9) where evaluated as demonstrating higher degrees of pleasantness confirms the logistic regression model that indicated IPI as the only significant predictor of pleasantness in BP: the higher the IPI, the more pleasant was the result, which is contrary to the proposal by Byers (1979) that more frequent pauses would be preferred.

For BP, a second model using the stretches of sound between two consecutive silent pauses as a predictor highlighted a preference for longer stretches of sound production, a result that does not hold for EP. It seems that Brazilian listeners, unlike their Portuguese counterparts, reject the positive aspect of expectation related to pause duration (Buraa 2019).

Thus, pause duration and rate are both relevant for predicting pleasantness, with the difference that Portuguese listeners seem to rely on pause duration and Brazilian listeners on pause rate (the inverse of IPI). These same preferences were already found in the first study, for the poem "Quando vier a primavera" (Barbosa 2022). Due to the differences found in the two varieties of Portuguese under study here, H6 (declamations with longer pauses are considered more pleasant) can be maintained only for EP, whereas H7 (declamations with lower rates of pauses are considered more pleasant) seems to be valid only for BP.

As for H5 (the main source of variability in terms of pause-related parameters is inter-reciter variation), results for effect size from mixed models confirm that poet and valence explained only up to 4% of variance for each dependent variable, whereas the participant explained from 5 to 12% of the total variance, individual values considered. The participant is, then, the main source of variability for pause duration and rate.

In the "dancing" of pauses within declamation, Brazilian listeners prefer reciters producing sonority, whereas Portuguese listeners prefer reciters pausing. Dancing is here understood as a mere metaphor, where different patterns of pauses (breaks) cause different aesthetical appreciations. This general picture suggests that the pleasantness of a recited poem is dependent on the culture of its reception, but the reasons for a particular choice are not easily inferred for at least two main reasons: (1) distinctions of formal education in different countries and/or cultures (European vs. South American); (2) the potential distinct effects on pleasantness of different syntactic and semantic breaks made by the reciters, an aspect that was not evaluated in the present work.

The results presented here, regardless of the reasons for particular levels of appreciation, have an impact on the choice of voices for TTS systems, and potentially for development of a more persuasive voice, because we have shown that changes in pausing affect pleasantness, which is certainly one of the elements of persuasion.

**Funding:** This research was funded by the Conselho Nacional de Desenvolvimento Científico e Tecnológico (CNPq), grant number 302194/2019-3, and the APC was funded by the same agency and grant.

**Institutional Review Board Statement:** The study was conducted in accordance with the Declaration of Helsinki, and approved by the Comitê de Ética em Pesquisa of the University of Campinas (protocol code 61430122.9.0000.8142 at 30 August 2022).

**Informed Consent Statement:** Informed consent was obtained from all subjects involved in the study.

**Data Availability Statement:** All stimuli are publicly available for listening at the following URL: https://figshare.com/authors/Plinio_Barbosa/11320902 (accessed on 1 October 2022). The audio files from the Current-Set are available in the folders "Pause study EP audio files" (for the declamations in EP), and "Pause study BP audio files" (for the declamations in BP). The declamations of "Quando vier a primavera", that constituted the previous study are in the folders "BP audio files" and "EP audio files".

**Acknowledgments:** The author thanks Alicia Crochiquia for preparing the online links for the perception tests for the two varieties of Portuguese; all persons that collaborate in finding participants for the perception tests and in finding the reciters. I am particularly in debt in this respect to Sandra Madureira for the Brazilian subjects, and Ana Rita Valente and Quintino Lopes for the Portuguese subjects (the three for help in finding both reciters and listeners in her/their own variety of Portuguese). I also thank all reciters and listeners from both sides of the Atlantic Ocean.

**Conflicts of Interest:** The author declares no conflict of interest. The funder had no role in the design of the study; in the collection, analyses, or interpretation of data; in the writing of the manuscript; or in the decision to publish the results.

**Appendix A**

Poems of the Current Set:

**O amor é uma companhia**
by Alberto Caeiro

O amor é uma companhia.
Já não sei andar só pelos caminhos,
Porque já não posso andar só.
Um pensamento visível faz-me andar mais depressa
E ver menos, e ao mesmo tempo gostar bem de ir vendo tudo.
Mesmo a ausência dela é uma coisa que está comigo.
E eu gosto tanto dela que não sei como a desejar.

Se a não vejo, imagino-a e sou forte como as árvores altas.
Mas se a vejo tremo, não sei o que é feito do que sinto na ausência dela.
Todo eu sou qualquer força que me abandona.
Toda a realidade olha para mim como um girassol com a cara dela no meio.

**Tempo**
by Adélia Prado

Enquanto eu fiquei alegre,
permaneceram um bule azul com um descascado no bico,
uma garrafa de pimenta pelo meio,
um latido e um céu limpidíssimo
com recém-feitas estrelas.
Resistiram nos seu lugares, em seus ofícios,
constituindo o mundo pra mim, anteparo
para o que foi um acometimento:
súbito é bom ter um corpo pra rir
e sacudir a cabeça. A vida é mais tempo
alegre do que triste. Melhor é ser.

**Momento**
by Adélia Prado

A mim que desde a infância venho vindo,
como se o meu destino,
fosse o exato destino de uma estrela,
apelam incríveis coisas:
pintar as unhas, descobrir a nuca,
piscar os olhos, beber.
Tomo o nome de Deus num vão.
Descobri que a seu tempo
vão me chorar e esquecer.
Vinte anos mais vinte é o que tenho,
mulher ocidental que se fosse homem,
amaria chamar-se Fliud Jonathan.
Neste exato momento do dia vinte de julho,
de mil novecentos e setenta e seis,
o céu é bruma, está frio, estou feia,
acabo de receber um beijo pelo correio.
Quarenta anos: não quero faca nem queijo.
Quero a fome.

Translation (corrected from a Google translation):

**Love is company**
by Alberto Caeiro

Love is company.
I no longer know how to walk alone on the paths
Because I cannot walk alone anymore.
A visible thought makes me walk faster
And see less, and at the same time enjoy seeing everything.
Even her absence is something that is with me.
And I like her so much that I do not know how to want her.

If I do not see her, I imagine her and I am strong as tall trees.
But, if I see her, I tremble, I do not know what is made of what I feel in her absence.
I am any force that abandons me.
All of reality looks at me like a sunflower with her face in the middle.

**Time**
by Adelia Prado

While I was happy,
remained a blue teapot with a peeling spout,
a bottle of pepper in the middle,
a bark and a crystal clear sky
with freshly created stars.
They resisted in their places, in their working tasks,
constituting the world for me, screen
for what was an affliction:
suddenly, it is good to have a body to laugh
and shake your head. Life is more
happy times than sad ones. It is better to be.

**Moment**
by Adelia Prado

I have been coming since childhood
as if my fate,
were the exact fate of a star,
calling for amazing things:
to paint your nails, to uncover the back of your neck,
to blink your eyes, to drink.
I take the name of God in a vault space.
I found out that, in time,
people will cry for me and forget me.
Twenty years plus twenty is what I have,
western woman who, if I were a man,
would love to be called Fluid Jonathan.
At this exact moment on the twentieth of July,
of year one thousand nine hundred and seventy-six,
the sky is haze, it is cold, I'm ugly,
I just got a kiss from the mail.
Forty years: I don't want a knife or cheese.
I want hunger.

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
