# Peer review of "The Dance of Pauses in Poetry Declamation"

_languages, doi:10.3390/languages8010076_

Round 1

Reviewer 1 Report

The article is devoted to a narrow task related to the field of river technologies. The topic of the article is relevant. The structure of the article does not correspond to the generally accepted one (Introduction, Models and methods, Experiments, Discussion, Conclusions). The level of English is acceptable. The article is too short. The text in figures 2, 5-9 is too small (like the figures themselves). The article cites mostly irrelevant sources.

The following remarks can be made on the material of the article:

1. I appreciate poetry (especially Omar Khayyam). At the same time, I am a scientist and scientists love precise definitions. In the title of the article, I was hooked by the term “Dance of Pauses”, but I did not find this “flight of butterflies” in the text of the article. The word "dance" generally occurs once - in the title. Thus, the authors wrote the article not about that.

2. Prosody is extremely important not only to make speech sound natural and lively, but also to get the most complete reflection of the speaker's speech style. In furnace technologies, prosodic speech features are represented as a four-dimensional vector per TTS unit (about one third of HMM sound states), including log-duration, start and end log-pitch, and log-energy. Why did the authors choose pauses for prosodic analysis - a priori, the least informative "particles" of speech. I absolutely do not understand. However, atovrs love statistics. In this case, I ask the authors to statistically justify that the description of prosodic based on the parametrization of speech pauses is the most informative. I allow using the criterion of relative entropy for justification.

3. In the abstract, the authors continue to intrigue my inner esthete. The term "degree of pleasantness". I ask the authors to show the “pleasantness” metric, justify its validity, and provide methods and algorithms for its calculation. And, of course, appreciate her place in Dance of Pauses.

4. Since the question of defining pauses in one form or another arises for almost everyone who works with sound, there are a lot of ways to solve it. The question of the necessary accuracy and the peculiarities of the audio material in each case determine how carefully the parameters must be selected, or otherwise you can limit yourself to a basic solution like YAAPT. Taking Praat for speech processing (nevertheless, a huge number of researchers use it), and not, for example, YIN, authors must justify their choice in the metric of the relevant criteria.

5. An absolute advantage of static analysis is the complete coverage of the analyzed set. The disadvantages of static analysis are the inevitable presence of false positives, resource consumption, and long scan times on large amounts of data. How did the authors overcome these shortcomings?

Author Response

I thank you each one of the reviewers for the time and care they took in reading the manuscript and presenting suggestions and comments. I addressed each one of them and modified the paper accordingly. All changes are in yellow and, in the case of rewording and of larger modifications, a marginal comment addressing the particular reviewers was added.

In the case of the figures, the major problem is not to see the reference to the subjects/participants in my opinion: that is why I enlarged the font. Thank you for pointing me that as well. In the case of Figure 9, I rotated the reference to the recitations.

In the cover letter, all reviews and my replies can be found and are preceded by "R." Here the replies to Reviewer # 1. Please, refer to the new version of the manuscript with changes in yellow and marginal notes commenting some of the changes.

Reviewer 2 Report

The present study The present study examines the role of pauses from a production and perception point of view in poem declamation. The research is professionally well founded, well thought out from a methodological point of view, the presentation of the results is detailed and logical, supported by statistical studies.

I only have a few minor critical comments and questions:

1. The reference is missing in lines 57, 123, 377; only (XXX YYYY) is mentioned.

2. There are too many hypotheses, H3 - H4 and H6 - H7 can be combined.

3. The axes labels of the figures are too small.

And finally, a question for conclusions:

What can explain the fact that Portuguese listeners seem to rely on pause duration and Brazilian listeners on pause rate in predicting pleasantness?

Author Response

I thank you each one of the reviewers for the time and care they took in reading the manuscript and presenting suggestions and comments. I addressed each one of them and modified the paper accordingly. All changes are in yellow and, in the case of rewording and of larger modifications, a marginal comment addressing the particular reviewers was added.

In the case of the figures, the major problem is not to see the reference to the subjects/participants in my opinion: that is why I enlarged the font. Thank you for pointing me that as well. In the case of Figure 9, I rotated the reference to the recitations.

In the cover letter, all reviews and my replies can be found and are preceded by "R." Here the replies to Reviewer # 2. Please, refer to the new version of the manuscript with changes in yellow and marginal notes commenting some of the changes.

Reviewer 3 Report

The submitted paper is a valuable contribution to the research field. There are a few minor points that could still improve it if addressed. They are listed below in the order in which they appear in the text.

p. 2, line 73: "genre in its own that not depends" - amend the grammar

p. 2, bottom: Since you are not giving any information on meter, number of syllables per line, phrasing, etc., you should refer the reader to the texts at the end of the article. It would eliminate the feeling of inadequate description of the material.

p. 4, top: Please provide the information about ordering of the test items. In what order did the listeners hear the individual rendering of of the poems? Randomized? Always the same? If the latter, did you use any desensitization sounds?

p. 4, lines 168-170: Could you explain how linear transformation of 1-2-3-4-5 to 0-25-50-75-100 helps the logistic model? It does not change the structure of the data, does it?

p. 5, line 212: "both the the negative poem" one extra "the"

p. 5, line 218: "the ten Portuguese female participants" - there are only five, not ten speakers in the graph

p. 7, line 238: "values replaces the individual values" the verb should be in plural

p. 7, line 252 and elsewhere: I agree that the Duncan Test is used for grouping but it would be more appropriate to talk about the resulting "sets" not "groups". This is because it is conventional to consider an empty set or a set with just one item in it. The "group" of one item does not sound right.

p. 7, line 261 and in many other legends to figure: "the last five ones from male reciter" - it should be "reciters" (I would actually suggest to reformulate the legend since the first five against the last five without anything in middle sounds a bit awakward. Why not five on the left against five on the right?)

p. 9, line 285: "in the end of verses" should be "at the end of verses"

p. 13, lines 359, 360: the word "here" is used twice, but it would be better to refer above to the actual sections, where the referents are.

p. 13, lines 368 to 373 and wordings elsewhere (e.g., p. 5, lines 210-213): These and similar wordings unnecessarily indicate a conceptual problem. It would be fair to point out that the pauses and IPIs are not affecting the listeners on their own, that is in isolation. There are effects caused by textual factors like correspondences of pauses and IPIs with syntactic and semantic structure. Those were not analysed, which is fine - it would be a huge task, but they should not be ignored. At least the discussion should mention them.

Author Response

I thank you each one of the reviewers for the time and care they took in reading the manuscript and presenting suggestions and comments. I addressed each one of them and modified the paper accordingly. All changes are in yellow and, in the case of rewording and of larger modifications, a marginal comment addressing the particular reviewers was added.

In the case of the figures, the major problem is not to see the reference to the subjects/participants in my opinion: that is why I enlarged the font. Thank you for pointing me that as well. In the case of Figure 9, I rotated the reference to the recitations.

In the cover letter, all reviews and my replies can be found and are preceded by "R." Here the replies to Reviewer # 3. Please, refer to the new version of the manuscript with changes in yellow and marginal notes commenting some of the changes.

Round 2

Reviewer 1 Report

I made the following recommendations to the basic version of the article:

1. I appreciate poetry (especially Omar Khayyam). At the same time, I am a scientist and scientists love precise definitions. In the title of the article, I was hooked by the term “Dance of Pauses”, but I did not find this “flight of butterflies” in the text of the article. The word "dance" generally occurs once - in the title. Thus, the authors wrote the article not about that.

2. Prosody is extremely important not only to make speech sound natural and lively, but also to get the most complete reflection of the speaker's speech style. In furnace technologies, prosodic speech features are represented as a four-dimensional vector per TTS unit (about one third of HMM sound states), including log-duration, start and end log-pitch, and log-energy. Why did the authors choose pauses for prosodic analysis - a priori, the least informative "particles" of speech. I absolutely do not understand. However, atovrs love statistics. In this case, I ask the authors to statistically justify that the description of prosodic based on the parametrization of speech pauses is the most informative. I allow using the criterion of relative entropy for justification.

3. In the abstract, the authors continue to intrigue my inner esthete. The term "degree of pleasantness". I ask the authors to show the “pleasantness” metric, justify its validity, and provide methods and algorithms for its calculation. And, of course, appreciate her place in Dance of Pauses.

4. Since the question of defining pauses in one form or another arises for almost everyone who works with sound, there are a lot of ways to solve it. The question of the necessary accuracy and the peculiarities of the audio material in each case determine how carefully the parameters must be selected, or otherwise you can limit yourself to a basic solution like YAAPT. Taking Praat for speech processing (nevertheless, a huge number of researchers use it), and not, for example, YIN, authors must justify their choice in the metric of the relevant criteria.

5. An absolute advantage of static analysis is the complete coverage of the analyzed set. The disadvantages of static analysis are the inevitable presence of false positives, resource consumption, and long scan times on large amounts of data. How did the authors overcome these shortcomings?

I did not like the answers of the authors with their subjectivity. Let me explain my point of view:

- "The reason is that previous research published with the evaluation of the poem "Quando vier a Primavera" showed that pausing (both rate and duration) have much more effect size that the other prosodic parameters analyzed and we decided to focus of pause due to that." Touches this "we decided". The scientist is based on data (results of measurements or surveys) and their consistent interpretation. For example, on many TV channels, a beautiful topless announcer reads the weather forecast. These programs are consistently popular. Maybe the reason is that the announcer effectively uses pauses during recitation? I think that the formulation of the study should be strengthened by arguments and facts.

- Pleasantness is evaluated by the lay listener with no further explanation. This is usual in studies of attractiveness and pleasantness in speech such as the ones by Rosenberg and Hirschberg (2020), Quené (2020), Strangert and Gustafson (2008), Hodges and Simeon (2010), Baumann (2017), Burkhardt et al . (2010), Weiss and Burkhardt, 2012, Weiss and Burkhardt (2010), and Carlsen et al. (2018)." Why "the lay listener with no further explanation"? Do the authors believe that an unprepared person will consider the paintings of the late Picasso to be art? This is wrong. I think that weighted assessment is optimal, when a person's opinion is multiplied by the weight coefficient of his professionalism. Moreover, unfortunately, poetry is not mainstream now.

- "Pauses were marked manually as defined by the absence of speech." One of the bases of science is the reproducibility of experimental results. This answer is denied by the authors. I ask you to either strongly argue for such an answer, or suggest another one.

- “The statistical methods used report effect sizes (Negelkerke pseudocorrelation) and 95% confidence intervals to evaluate the impact of the predictors on the predicted variables. We do not rely on p values only or significant decisions. Furthermore, I now added an inter-rater-reliability test to discuss the coherence between the responses.” This interpretation of the statistics looks touching. Let's just use the appropriate statistical criteria to justify the validity of the experimental results and the representativeness of the data?

Author Response

Addressed to editors.